# The Consequences of the COVID-19 Pandemic on Emergency Surgery for Colorectal Cancer

**DOI:** 10.3390/ijerph20032093

**Published:** 2023-01-23

**Authors:** Catalin Vladut Ionut Feier, Sonia Ratiu, Calin Muntean, Sorin Olariu

**Affiliations:** 1First Discipline of Surgery, Department X-Surgery, “Victor Babes” University of Medicine and Pharmacy, 2 E. Murgu Sq., 300041 Timisoara, Romania; 2First Surgery Clinic, “Pius Brinzeu” Clinical Emergency Hospital, 300723 Timisoara, Romania; 3Plastic Surgery Clinic, “Casa Austria, Pius Brinzeu” Clinical Emergency Hospital, 300723 Timisoara, Romania; 4Department of Informatics and Medical Biostatistics, “Victor Babes” University and Pharmacy, 300173 Timisoara, Romania

**Keywords:** colorectal cancer, emergency surgery, COVID-19 pandemic, severe symptoms, postoperative mortality, stoma protection

## Abstract

The aim of this study is to analyze the impact of the COVID-19 pandemic on the emergency treatment of patients with colorectal cancer in a university surgery clinic. Data from patients undergoing emergency surgery during the pandemic period (2020–2021) was taken into consideration and the results were analyzed and compared with the periods 2016–2017 and 2018–2019. A significant decrease in the number of patients undergoing emergency surgery was reported (*p* = 0.028). The proportion of patients who presented more severe symptoms at the hospital was significantly higher (*p* = 0.007). There was an increase in the average duration of surgical interventions compared to pre-pandemic periods (*p* = 0.021). An increase in the percentage of stomas performed during the pandemic was reported. The average duration of postoperative hospitalization was shorter during the pandemic. A postoperative mortality of 25.7% was highlighted. Conclusions: The pandemic generated by COVID-19 had significant consequences on the emergency treatment of patients with colon cancer. A smaller number of patients showed up at the hospital, and with more severe symptoms. In order to reduce the risk of infection with SARS-CoV-2 virus, the postoperative hospitalization period was shortened and a higher number of protective stomas were performed.

## 1. Introduction

The COVID-19 pandemic had important consequences on emergency surgical treatment for colorectal cancer all over the world. The first person infected with the SARS-CoV-2 virus in Romania was reported on 26 February 2020, and the state of emergency was introduced on 16 March 2020. The situation had escalated exponentially, with a daily increase of the number of COVID 19 cases. Healthcare systems were taken by surprise worldwide, and the first measures taken involved the redistribution of material and human medical resources to increase the support of intensive care units for the management of infected patients with the new virus [1].

Colorectal cancer management has been severely impacted by the pandemic. This pathology represents the third most common cancer pathology worldwide [2]. The screening of this pathology was severely reduced during the pandemic, with many cases remaining undiagnosed. In addition to this, the initial directives were to postpone elective surgical interventions [3]. This led to an increase in the number of patients presenting to the emergency room with complications of colorectal cancer (complete large bowel obstruction or partial bowel obstruction, lower digestive hemorrhage, perforation of the colon with generalized peritonitis) and a much more severe symptomatology, leading to an increased risk of mortality and morbidity [4]. The purpose of this study was to highlight the consequences of the pandemic on the emergency treatment of colorectal cancer and the management of patients with this pathology. The impact on the condition of the patients upon presentation to the hospital, their evolution, and the outcome were monitored as well.

## 2. Materials and Methods

This study includes patients who underwent emergency surgery for the treatment of colon cancer in the First General Surgery Clinic of the Timisoara County Emergency Clinic Hospital, Pius Brinzeu (one of the hospital’s three surgery clinics). 

The study included patients admitted in the timeframes: 26 February 2020 to 1 October 2021 in comparison to the same interval for 2016–2017 and 2018–2019. There were multiple inclusion criteria: patients undergoing emergency surgery for colorectal cancer in the previously mentioned periods with a primary tumor whose location varies from the level of the cecum to the level of the rectosigmoid junction.

During the pandemic, additional inclusion criteria were added, such as the absence of COVID-19 infection upon presentation to the emergency service, or the history of infection with the SARS-CoV-2 virus, as well as the absence of any symptoms specific to SARS-CoV-2 infection with the new coronavirus 7 days prior to presentation to the hospital. Lastly, only patients who presented a negative result in the RT-PCR test performed at the time of hospitalization were taken into consideration. 

The exclusion criteria were represented by patients who underwent surgery outside the mentioned time interval, patients who underwent elective surgery for the treatment of colorectal cancer, and a location of the primary tumor at a different level of the colon than the one mentioned previously. 

During the pandemic, additional exclusion criteria were added, such as the presence of infection with the SARS-CoV-2 virus at the time of hospitalization, or a history of infection as well as the presence of infection during hospitalization with this virus. Moreover, other exclusion criteria were represented by the presence of any specific COVID-19 symptoms at the time of presentation to the hospital or 7 days before presentation, as well as obtaining a positive result in the RT-PCR test for the detection of COVID-19. 

Data collection was carried out after obtaining the Ethical Approval of the Hospital Commission (NO. 319/04.08.2022). 

Multiple parameters were investigated, such as date of birth, gender, age, environment origin, date of admission, date of discharge, total number of hospitalization days, total number of postoperative hospitalization days, duration of the surgical intervention, and the presence of severe symptoms (such as abdominal pain, emesis, reduced flatulence and hemorrhage of the lower digestive tract). Associated pathologies of the patients were evaluated with the help of the Charlson index. The presence of complete large bowel obstruction and partial bowel obstruction at the time of surgery was also analyzed, as well as the location of the tumor (ranging from the cecum, ascending colon, hepatic angle, transverse colon, splenic angle, descending colon, and rectosigmoid junction). Furthermore, the type of surgical intervention performed (right colectomy, left colectomy-left hemicolectomy or Hartmann procedure, transverse colon segmental resection, and other colectomies) was noted. The presence of recurrence, the type of intervention (curative/palliative) as well as whether mechanical or wire anastomosis was performed and the frequency of protective stomas were analyzed. The need for postoperative transfusions, the need for monitoring in intensive care units, and the presence of fistula as a postoperative complication were taken into consideration. In addition to these, the degree of tumor invasion (T), the number of lymph nodes with metastases (N), as well as the presence and absence of metastases (M) along with lympho-vascular invasion and the stage of the disease were analyzed. Finally, the discharge status of the patients was analyzed (cured, improved, stationary, aggravated, deceased). 

The IBM SPSS Statistics (IBM, Armonk, NY, USA) for Windows program was used to analyze the collected data. The determination of the average mode and the median was carried out for the numerical variables that presented a normal distribution. Regarding the categorical variables, they were analyzed based on frequency tables and percentages. Statistical tests for multiple samples (ANOVA) or Chi-Square proportion tests were applied with a *p* < 0.05 being considered statistically significant.

## 3. Results

In order to carry out this study, the data of patients undergoing emergency surgery for the treatment of colorectal cancer during the period 26 February 2020 to 01 October 2021 was analyzed and compared to the same period for the years 2016–2017 and 2018–2019. 

The study was carried out on a group of 142 patients that met the inclusion criteria. A total of 35 surgeries (24.6%) were performed during the pandemic, 59 surgeries (41.5%) were performed in the 2018–2019 timeframe, and the other 48 surgeries (33.8%) were conducted in the 2016–2017 period. Following the statistical analysis after applying the Chi-square test to study the proportions between the three periods resulted in a *p* = 0.028. 

The average age of the patients in the study was 67.99 years (min.26–max.97), SD = 12.29 and a Median = 70. Of these, 51.4% were men, 48.6% were women, and the majority of cases came from the urban environment (59.9%).

The presence of severe symptoms at admission is presented in Table 1.

Patients’ comorbidities were quantified using the Charlson index. In the first period of the study, the average index was 4.23 (min.2–max.8, SD 1.86), with an average of 3.69 (min.2–max.10, SD 1.71) during 2018-2019 and an average of 3.91 (min.2–max.9, SD 1.95) for the pandemic period.

The presence of incomplete or complete ileus was reported in 38 cases (79.2%) in the first period of the study, and in 51 cases (86.5%) in the 2018–2019 period. During the pandemic, out of 35 patients, 22 (62.9%) presented complete large bowel obstruction and nine (25.7%) presented partial bowel obstruction. After studying the difference in proportions between the three periods, a *p* = 0.029 was obtained.

The tumor location and type of surgery used are presented in Table 2.

During the pandemic, four patients (11.4%) presented recurrent disease, compared to the 2016–2017 period which saw of two patients (4.2%) with recurrent disease, and the 2018–2019 period, which saw five cases of recurrent disease(8.5%). Curative treatment was also applied in 28 cases (80%) during the pandemic, compared to 84.7% of cases in the 2018–2019 period. 

The anastomosis was created in 60.4% of the cases during 2016–2017 and all of them were done manually. During 2018–2019, the anastomosis was performed in 45.8% of cases, and in all cases it was performed manually. During the pandemic, the percentage of anastomoses performed was 42.9%, and in three patients mechanical anastomosis with a stapler was applied. 

The protective stoma was made in 39.6% of cases in the first period of the study, and increased to 57.1% during the pandemic. This means that 20 patients ended up with a protective stoma during the pandemic. Out of the 18 patients who underwent left colectomy, 14 ended up with the Hartmann procedure and implicitly with a protective stoma, while in the other four situations a left hemicolectomy was performed. All six patients who underwent other colectomies ended up with a protective stoma, since they required more extensive and laborious surgical acts. 

The need for postoperative transfusions was also analyzed. Thus, during the pandemic, 17 patients (48.6%) required transfusions, compared to 21 patients (35.6%) in the 2018–2019 period, and 22 patients (45.8%) in the 2016–2017 period. 

The presence of intestinal fistula as a postoperative complication was reported in five cases (10.4%) in the 2016–2017 period, in two cases (3.4%) in the second period of the study, and in no cases during the pandemic period. 

The necessity of monitoring patients in the postoperative intensive care units was the highest in the first period of the study, with a percentage of 29.2%, followed by the 2018–2019 period with a percentage of 20.3%, and 17.1% during the pandemic.

The T (site and size of primary tumor), N (regional lymph node involvement) M (presence of metastatic spread) variation, presence of lympho-vascular invasion and the cancer stage throughout the three periods are presented in Table 3

In order to study the effect of the pandemic on the emergency treatment of colorectal cancer, parameters such as the number of hospitalization days, the duration of the surgical intervention and the time spent by the patients in the hospital after the intervention were analyzed. 

Thus, in the period 2016–2017, the duration of the surgical intervention was on average 174.77 min, (min.60–max.285, SD.57.72). In the second period of the study, the average duration was 199.66 min (min.60–max.510, SD.78.54), the longest duration of the intervention being reported during the pandemic with an average of 217.38 min (min.80–max.360). Following the application of statistical tests to highlight the differences between the three periods, a *p* = 0.021 was obtained. 

The number of postoperative hospitalization days was on average 12.27 (min.1–max.24, SD.5.2) in the period 2016–2017, in the second period the average was 12.46 (min.1–max.50, SD.9.42), and during the pandemic this average dropped to 11.14 days (min.1–max.32, SD.5.49). 

The average length of hospitalization was 14.38 days in the 2016–2017 period (min.1–max.36, SD, 6.01), in the period 2018–2019 the average was 14.08 days (min.1–max.56, SD, 10.47), and during the pandemic, it dropped to 12.63 days (min.1–max.35, SD,6.32).

The discharge status of the patients also varied throughout the three periods. The number of patients who died postoperatively in the period 2016–2017 was 11 (22.9%), in the second period of the study this percentage dropped to 16.9%, and during the pandemic it rose to 25.7%, with 9 of the 35 patients dying. 

After studying the correlations and associations between the variables presented above, the following results were obtained during the pandemic:

During the pandemic, an inverse correlation between the age of the patients and the duration of the intervention (r = −0.355, *p* = 0.039) was highlighted. 

The total number of days of hospitalization correlates positively with the Charlson index (r = 0.460, *p* = 0.039) and with the number of days of postoperative hospitalization (r = 0.928, *p* < 0.001). 

Following the statistical analysis, there was an association between the female gender and the presence of intestinal occlusion (*p* = 0.033). On the other hand, an association between the male gender and the need for transfusions was highlighted (*p* = 0.031). 

The association between weight loss of patients and tumor location was presented (*p* = 0.026), patients without weight loss had left-sided colon cancer (66.7%), and those with weight loss had right-sided colon cancer (80%). 

Age is associated with the presence of a protective stoma (*p* = 0.034). The average age of the patients who had a protective stoma was 70.11 years, compared to 65.75 years for those without. 

The patients who required post-operative monitoring in the intensive care units presented an average age of 72.03 years compared to 66.82 years for those who were not hospitalized in the intensive care units, following the application of statistical tests a *p* = 0.034 resulted. 

The discharge status of the patients is associated with the type of intervention (curative/palliative), and following the statistical analysis a *p* = 0.05 was obtained. Out of those who benefited from a curative treatment, 78.6% were discharged cured, compared to only 42.9% of those who underwent a palliative surgical treatment. Regarding the association between discharge status and the presence of recurrent disease, an association was detected between these variables (*p* = 0.018). Among the patients who presented this characteristic, 50% died postoperatively. 

Furthermore, an association between the discharge status of the patients and the need for monitoring in intensive postoperative therapy (*p* = 0.041) was highlighted. A total of 66.7% of the patients who required monitoring in the intensive care clinic died postoperatively. 

Following the analysis, an association between the need for postoperative transfusions and the stage of the patients’ disease (*p* = 0.014) was determined. Thus, 37.5% of these patients were at stage IV, and 37.5% were at stage II.

## 4. Discussion

The pandemic required a series of directives from the healthcare systems around the world in order to be able to deal with the increasing number of infections. These healthcare systems focused mainly on the management of patients infected with the SARS-CoV-2 virus, and as a result, oncological pathologies were neglected in the first phases of the pandemic. 

The emergency surgical treatment of colorectal cancer patients during the pandemic was severely affected. Therefore, 40.67% fewer patients underwent emergency surgery compared to the 2018–2019 period, and there were 27.08% fewer interventions compared to the 2016–2017 period. Thus, applying the Chi-square test, a *p* = 0.028 was obtained, which shows that there are significant differences between the proportion of those undergoing emergency surgery during the pandemic and the rest of the periods. During the pandemic, the number and proportion of those undergoing emergency intervention is significantly lower compared to previous periods. This is due to the restrictions imposed by the authorities and their indications not to visit hospitals except in the case of severe symptoms. Moreover, patients feared encountering a potentially infected person with the SARS-CoV-2 virus and drastically reduced their presentation to the emergency service.

Finally, the epidemiological procedures and the increased waiting time required to undergo an examination have led to an increase in the reluctance of patients to visit hospitals. These aspects were also mentioned in a study carried out in Denmark, which shows a 48% decrease in the number of emergency surgeries performed during the pandemic, and in Scotland with a decrease of 58% [5,6]. In the USA, this percentage dropped by a maximum of 61% during the pandemic [7]. 

Following the analysis of the proportions of patients who presented severe symptoms at the time of presentation to the emergency service, a *p* = 0.007 was achieved, resulting in very significant differences between the three periods. 

During the pandemic, 94.3% of patients presented severe symptoms compared to approximately 66% in the other two periods. This is due to the postponement of the visit to the hospital until the appearance of unbearable symptoms, the restrictions and indications of the authorities, as well as the patients’ fear of coming into contact with possibly infected people. 

The literature also supports these ideas, showing an extremely significant difference between the pandemic period and the pre-pandemic period in terms of the number of patients who presented severe symptoms [5]. In addition, some studies show an increase of up to 20.9% in patients who presented severe symptoms [8]. Studies were highlighted showing that the proportion of patients with severe symptoms was significantly higher during the pandemic (*p* < 0.001) [9].

Furthermore, due to the massive reduction in colorectal cancer screening patients who were unaware of their pathology, and failed to associate the presence of mild symptoms with the presence of this pathology, only severe symptoms caused them to decide to visit the hospital [7,10,11,12,13].

Undesirable complications of cancer can be represented by the presence of complete large bowel obstruction and partial bowel obstruction. The presence of these two was significantly higher during the pandemic, following the application of statistical tests resulting in a *p* = 0.029. This aspect is due to the postponement of the visit to the hospital, and to a more advanced stage of the disease. A study carried out in a hospital in Bucharest shows a significant increase in the number of patients who presented intestinal occlusion during the pandemic compared to the pre-pandemic period (*p* = 0.01) [14]. In another study, a twofold increase in the percentage of patients presenting with large bowel obstructions was reported in 2020 compared to 2018–2019 (*p* = 0.01) [15]. 

Furthermore, the percentage of patients with recurrent disease was higher during the pandemic, although in absolute value during 2018–2019 there were five patients who presented recurrent disease; as a percentage they have a lower proportion (8.5%) compared to 11.4% during the pandemic. 

Due to the need to reduce the risk of contacting the SARS-Cov-2 virus and also a shorter hospital stay, 57.1% of the patients underwent a protective stoma during the pandemic compared to 39.6% in the first period of the study. An unscheduled increase in protective stomas during the pandemic was reported in the literature as well [9]. Another study shows a doubling in the number of patients with a protective stoma, which was done to reduce postoperative complications [16,17].

An important aspect is represented by the absence of intestinal fistula as a complication during the pandemic. It should be mentioned that this parameter is also directly related to an increased number of stomas made and implicitly to a lower number of anastomoses. A similar incidence of the frequency of postoperative complications was reported both in the pandemic and in the pre-pandemic period in a study (*p* = 0.118) [18].

The proportion of patients requiring postoperative intensive care monitoring was 29.2% during the pandemic, an increase by 43.84% compared to the 2018–2019 period, and 70.76% compared to the first period of the study. This is due to the presentation of patients with a more advanced stage, with more severe symptoms, patients with changes in the complete blood count, biochemistry analyses and hydro electrolytic disorders. In specialized literature, an increase from 16.4% to 36.1% in the number of patients who required postoperative monitoring in intensive care units during the pandemic was reported [19]. 

According to the Table 3, a significant increase of patients with stage IV compared to the 2018–2019 period was seen, and the absence of patients with stage I can be observed. This can be explained by postponing the patients’ visit to the hospital until the last moment, and patients with stage I not presenting severe symptomatology, along with the fear of visiting hospitals; furthermore, due to the imposed restrictions, they did not become surgical emergencies at the time. 

Due to the reduced amount of screening during the pandemic, many patients were not diagnosed with stage I colon cancer. This aspect was also described in the literature, where a significant change in stage N (*p* = 0.024) and M (*p* = 0.018) was reported, which means that more advanced cancer presentations occurred during the pandemic [18].

Following the application of the Anova test, there were statistically significant differences between the three periods regarding the duration of the surgical intervention, the longest interventions taking place during the pandemic. This is due to the epidemiological procedures and rules that had to be followed to prevent the spread of the SARS-CoV-2 virus, including a more careful isolation of the operating field, the use of additional protective equipment, as well as the more complex interventions needed in order to treat patients with a more advanced pathology. 

With the risk of infection with the new coronavirus being higher in healthcare facilities, this led to shorter postoperative hospitalizations, with an average of this duration lower than that found in the first two periods of the study. 

There are studies that show a significant decrease in the duration of hospitalization, with an average of 5 days [17]. In a study carried out by Queens Hospital Burton, a decrease from 8 days to 5 days of hospitalization during the pandemic was reported [20]. Other studies carried out in Sao Paolo and Madrid also show a reduction in the period of hospitalization during the pandemic, with an average hospitalization of 11.7 days [21,22].

Discharge status was also influenced by the pandemic. Although in the second period of the study the postoperative mortality was 16.9%, during the pandemic this percentage rose to 25.7%. According to some conservative estimates made by a study from the USA, 4000 additional deaths are expected by the year 2030 due to the impact of the pandemic on the treatment of colon cancer [23,24]. Furthermore, another study from the United Kingdom estimated that an approximately 5% and 16% increase in deaths are likely to occur from colorectal cancer due to delay in diagnosis and therapeutic interventions, respectively [16,25].

Thus, an inverse correlation was observed between the patient’s age and the duration of the surgical intervention (r = −0.355, *p* = 0.039), suggesting that the intervention was shorter in older patients. This is due to the limitation of the surgical act to the strictly necessary maneuvers, in order not to subject the patient to a more extensive intervention than necessary. The purpose was to undertake only necessary surgical interventions in order to reduce recovery periods and obviously the risk of postoperative complications. This is also due to the association between the age of the patients and the performance of a protective stoma (*p* = 0.034); the average age of the patients who benefited from a protective stoma being higher than the others. This procedure implies a shorter duration of the intervention, and a faster postoperative recovery, avoiding hospitalization in intensive care units for patients, as well as a reduced impact on the patients. 

Age was a parameter that influenced the need for monitoring in the intensive care unit. Thus, an association between age and the necessity of postoperative intensive care monitoring was highlighted (*p* = 0.034), with an average age of 72.03 years compared to those who did not, who had an average age of 66.82 years. An obvious aspect was also emphasized in this study. Thus, patients who underwent palliative surgical treatment (stage IV patients) died postoperatively at a percentage of 57.1% (*p* = 0.05), compared to only 22.4% of those who underwent curative treatment. According to the literature, significantly more stage IV patients presented during the pandemic compared to previous periods [2]. 

Finally, the need for postoperative transfusions was associated with the stage of the disease, and thus 37.5% of the patients who required transfusions were stage IV, and the same percentage were stage II. This is due to the postponement of the initial visit to a hospital due to the restrictions, as well as the fear of visiting a hospital. Patients presented with a more advanced stage of the disease and an additional degree of weakness, with a precarious biological status. 

## 5. Conclusions

The COVID-19 pandemic had an important impact on health care systems worldwide. This study, alongside the existing literature, describes a dramatic decrease in the total number of surgeries performed for the treatment of colorectal cancer during the pandemic. 

It is a well-known fact that surgical resection remains the main treatment for colorectal cancer. During this period, the patients who underwent surgical treatment presented at the hospital with a more advanced stage of cancer and more severe clinical manifestations. 

In order to cope with the risk of SARS-CoV-2 infection, patients spent a shorter period in the hospital, and a higher proportion of stomas were performed. Our single center research does not exclude the possibility of a higher number of patients that were not diagnosed during this period, and that could lead to a higher mortality rate in the following years. This study shows that postponing the surgical intervention and a delay in patients presentation to hospital have important consequences on their outcome. Therefore, healthcare systems should focus on creating special care units for contagious patients oppositely to reorganizing surgical-oncological departments during such crisis periods. 

## Figures and Tables

**Table 1 ijerph-20-02093-t001:** Presence of severe symptomatology.

Symptomatology	2016–2017	2018–2019	2020–2021
Mild	16 (33.33%)	19 (32.2%)	2 (5.7%)
Severe	32 (66.7%)	40 (67.8%)	33 (94.3%)
After Chi square test: *p* = 0.007 between all 3 periods compared			

**Table 2 ijerph-20-02093-t002:** Tumor location and type of surgery performed.

		2016–2017	2018–2019	2020–2021
Tumor location	Right colon (%)	18 (37.5%)	13 (22%)	10 (28.6%)
	Left colon	28 (58.3%)	38 (64.4%)	20 (57.1%)
	Transverse colon	1 (2.1%)	8 (13.6%)	4 (11.4%)
Intervention type	Right colectomy	18 (37.5%)	14 (23.7%)	10 (28.6%)
	Left colectomy	23 (47.9%)	32 (54.2%)	18 (51.4%)
	Segmental resection of transverse colon	1 (2.1%)	5 (8.5%)	1 (2.9%)
	Other colectomies	6 (12.5%)	8 (13.6%)	6 (17.2%)

**Table 3 ijerph-20-02093-t003:** TNM variation, presence of lympho-vascular invasion and the stage during the three periods of the study.

		2016–2017	2018–2019	2020–2021
TNM	T1	-	1 (2%)	-
	T2	1 (2.3%)	4 (5.9%)	-
	T3	24 (50%)	26 (43.1%)	10 (30.3%)
	T4a	19 (38.6%)	24 (41.2%)	14 (42.4%)
	T4b	4 (9.1%)	4 (7.8%)	9 (27.3%)
	N0	12 (25%)	24 (41.2%)	12 (34.28%)
	N1	22 (45.83%)	22 (37.28%)	15 (42.85%)
	N2	14 (29.16%)	13 (22.03%)	8 (22.8%)
	M0	35 (72.7%)	52 (88.13%)	27 (77.14%)
	M1	13 (27.3%)	7 (11.86%)	8 (22.85%)
Lympho-vascular invasion	YES	28 (59.1%)	26 (44.06%)	21 (60.5%)
	NO	20 (33.89%)	33 (55.93%)	14 (39.4%)
Stage	I	2 (4.16%)	3 (5%)	-
	II	11 (22.91%)	25 (42.37%)	10 (28.57%)
	III	23 (47.91%)	24 (40.67%)	17 (48.57%)
	IV	12 (25%)	7 (11.86%)	8 (22.85%)

## Data Availability

The datasets used and/or analyzed during the current study are available from the corresponding author on reasonable request.

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
