# Peer review of "The Consequences of the COVID-19 Pandemic on Emergency Surgery for Colorectal Cancer"

_ijerph, 2023, doi:10.3390/ijerph20032093_

Round 1

Reviewer 1 Report

This is a retrospective single center review of patients undergoing surgery for colon cancer during the pandemic and comparison with two similar time periods before the pandemic.  It is mainly a descriptive study.  Because the patient numbers are understandably small, the number of actual statistically significant results is limited.  However the results are likely representative of the general experience during the pandemic and the relevance for this special issue of the journal is high.

For the most part, the English grammar is good.  However the writing is somewhat long and windy and should be condensed.  It should be more focused on those results that are actually statistically significant.  There are many areas of redundancy that could be shortened.  Sentences like this “After studying the difference in proportions between the 3 periods, a p=0.029 was obtained between the 3 periods” could be changed to just “(p=0.029)”.

Author Response

Firstly, I would like to thank you very much for you time and for reviewing our article. I have taken into considerations your requests and I have made modifications accordingly. Even if there are results that are not statistically significant (some of them maybe due to the small number of patients included as you said), I think they should be mentioned in order to expose as many consequences as possible this pandemic had on these patients.

Kind regards,

Sonia Ratiu

Reviewer 2 Report

First of all, I want to congratulate the authors for their work.

Here are my comments:

1) Medical English editing of the manuscript is required.

2)  The title of the article is “The consequences of the Covid-19 pandemic on emergency surgery for colon cancer”. Surgical interventions in colorectal surgery are well known.

The authors should mention in details the type of operation in study group from pandemic period 2020-2021 (eg. number of cases with Hartmann procedure) as it is written that percentage of anastomoses was only 42,9%.

3) Page 3 line 123 line 124

The authors should mention number of cases with complete large bowel obstruction and partial bowel obstruction.

Author Response

Firstly, I would like to thank you very much for you time and for reviewing our article.

Point 1. I have taken into considerations your request and I have made modifications accordingly.

Point 2. I understand that colorectal cancer is a more well-known medical term, and I changed throughout the document the term ,,colon cancer’’ with ,, colorectal cancer’’. I also presented  the surgical interventions patients underwent during the pandemic period, explaining the percentage of protective stomas as well.

Point 3. I have mentioned how many patients had complete or incomplete bowel obstruction during the pandemic.

I hope our modifications fulfill your requests.

Kind regards,

Sonia Ratiu